# Control of Tumor Progression by Angiocrine Factors

**DOI:** 10.3390/cancers13112610

**Published:** 2021-05-26

**Authors:** Elisenda Alsina-Sanchis, Ronja Mülfarth, Andreas Fischer

**Affiliations:** 1Division Vascular Signaling and Cancer, German Cancer Research Center (DKFZ), 69120 Heidelberg, Germany; e.alsina@dkfz.de (E.A.-S.); r.muelfarth@dkfz.de (R.M.); 2Faculty of Biosciences, University of Heidelberg, 69120 Heidelberg, Germany; 3Department of Internal Medicine, Endocrinology and Clinical Chemistry, University of Heidelberg, 69120 Heidelberg, Germany; 4European Center for Angioscience (ECAS), Medical Faculty Mannheim, University of Heidelberg, 68167 Mannheim, Germany

**Keywords:** angiocrine, blood vessels, endothelial cells, cancer, tumor progression, metastasis

## Abstract

**Simple Summary:**

The growth of a solid malignant tumor mass depends on the formation of new blood vessels by endothelial cells. However, endothelial cells do not only provide conduits for blood transportation, but also express numerous factors which promote the aggressiveness of cancer cells, influence the immune response toward cancer cells and thereby contribute to tumor progression and metastasis. This Review provides a comprehensive overview about such angiocrine factors and how they orchestrate the tumor microenvironment.

**Abstract:**

Tumor progression, therapy resistance and metastasis are profoundly controlled by the tumor microenvironment. The contribution of endothelial cells to tumor progression was initially only attributed to the formation of new blood vessels (angiogenesis). Research in the last decade has revealed however that endothelial cells control their microenvironment through the expression of membrane-bound and secreted factors. Such angiocrine functions are frequently hijacked by cancer cells, which deregulate the signaling pathways controlling the expression of angiocrine factors. Here, we review the crosstalk between cancer cells and endothelial cells and how this contributes to the cancer stem cell phenotype, epithelial to mesenchymal transition, immunosuppression, remodeling of the extracellular matrix and intravasation of cancer cells into the bloodstream. We also address the long-distance crosstalk of a primary tumor with endothelial cells at the pre-metastatic niche and how this contributes to metastasis.

## 1. Introduction

A solid tumor mass consists not only of cancer cells, but of numerous other resident and infiltrating cells and the extracellular matrix, which together form the tumor microenvironment (TME). The TME contains three main cell entities: fibroblasts, immune cells and endothelial cells. It is widely accepted that the TME supports the survival of cancer cells and is crucial for tumor progression and metastasis [1], and consequently there is increasing interest in targeting the tumor stroma to improve the effectivity of cancer therapies [2]. However, the complex interplay between cancer cells and the surrounding stroma is still incompletely understood.

In recent years the understanding of the role of endothelial cells (ECs) within the tumor stroma has substantially changed. ECs can no longer be seen as simple building blocks for new blood vessels (tumor angiogenesis), which nourish the tumor mass. Instead, ECs play crucial roles in facilitating tumor growth, providing a large platform of membrane-bound proteins as well as secreted factors interacting with cancer cells and other cells of the TME. These perfusion-independent functions are referred to as “angiocrine functions” [3]. As such, ECs orchestrate several aspects of cancer progression and metastasis through angiocrine functions [4,5,6,7]. Notably, such angiocrine functions play essential roles during development, organ regeneration and maintenance of organ homeostasis [3,8]. Therefore, it appears very likely that cancer cells hijack such physiological programs to generate a milieu that facilitates tumor growth. In this Review we summarize and discuss the communication of ECs with other cell types within the TME through the membrane-bound and angiocrine secreted factors and how this affects tumor progression.

## 2. Tumor Angiogenesis

The limited diffusion distance of oxygen requires that almost every cell of the body is within 100 to 150 µm of a capillary [9]. Therefore, the growth of a solid tumor usually requires the growth of new blood vessels from pre-existing ones [10]. This is achieved by the re-activation of the quiescent resident vasculature by growth factors such as the vascular endothelial growth factor (VEGF) secreted from cells within a hypoxic tissue. In a physiological setting, the growth of new blood vessels would lead to oxygen and nutrient delivery resulting in a reduced secretion of pro-angiogenic factors and the adoption of a quiescent vascular phenotype. In tumors, however, there is often persistent vascular growth factor secretion (e.g., due to mutations in cancer cells or by certain immune cells), and this results in aberrant angiogenesis, leading to a chaotically structured vasculature with impaired perfusion, cellular junctions integrity and poor coverage with mural cells (pericytes and vascular smooth muscle cells). This poorly functional vasculature further promotes hypoxia, immunosuppression and thereby tumor progression [11]. Here, we do not intend to further discuss tumor angiogenesis, as this has been extensively done by others [12,13,14]. However, it is important to mention that the immature tumor vasculature, lacking proper coverage with mural cells, provides a large signaling platform which, besides the release of soluble angiocrine factors, also enables ligand-receptor interactions of membrane-bound proteins between ECs and cancer cells or other cells of the TME.

We would like to mention only briefly that, based on the concept of tumor angiogenesis [15], anti-angiogenic cancer therapy has been developed and is nowadays the standard care in several tumor entities [16]. Despite the promising and impressive efficacy in prolonging progression-free survival, there is limited impact on overall survival. This might be due to acquired resistance mechanisms, such as the recruitment of alternative angiogenic pathways [17,18]. Very recently, fascinating data were reported showing that the combination of immunotherapy with anti-angiogenic treatment improves the outcome in patients with unresectable hepatocellular carcinoma [11,19]. The phase 3 IMBrave trial investigated a treatment with antibodies against VEGF (Bevacizumab) and PD-L1 (Atezolizumab) against the standard treatment with the tyrosine kinase inhibitor Sorafenib. The overall survival at 12 months was 67.2% (95% CI, 61.3 to 73.1) with Atezolizumab-Bevacizumab and 54.6% (95% CI, 45.2 to 64.0) with Sorafenib [19]. This promising finding demonstrates that there are important functions of ECs that go beyond the formation of new blood vessels. In the following chapters we will discuss these with a focus on angiocrine functions.

## 3. Beyond Angiogenesis

Under physiological conditions, ECs coordinate organ development, regeneration and homeostasis by angiocrine factors in an organ-specific manner [3,8,9]. There is increasing evidence that cancer cells can take advantage of such functions to generate a microenvironment that promotes tumor progression. We will first summarize the angiocrine factors which play a pivotal role in tumor models and subsequently discuss their roles in several aspects of tumor progression, immunosuppression and metastasis.

### 3.1. Angiocrine Factors

A number of soluble and membrane-bound angiocrine factors that influence tumor progression through action on the cancer cells themselves or on the local immune cells in the TME were discovered in the last years (Table 1). Several of these can be grouped into: (i) EC adhesion proteins, like intercellular adhesion molecule 1 (ICAM1), vascular cell adhesion molecule 1 (VCAM1), E-selectin and P-selectin, which are involved in the recruitment of leukocytes but also in the transmigration of cancer cells across the vessel wall; and (ii) chemokines, like interleukin-8 (IL-8 also known as CXCL8), monocyte chemotactic protein 1 (MCP1; also known as CCL2), stromal cell-derived factor 1 (SDF1; also known as CXCL12) and other factors which influence the recruitment and polarization of immune cells [3,6].

### 3.2. Angiocrine Control of Cancer Progression

Cancer progression is a complex process involving cancer cell proliferation, acquisition of different cancer cell phenotypes, such as stemness or epithelial to mesenchymal transition (EMT), degradation of the extracellular matrix (ECM), tissue invasion, intravasation, extravasation, immunosuppression (Section 3.3) and formation of metastases. All of these processes are influenced by angiocrine factors (Figure 1).

#### 3.2.1. Cancer Cell Proliferation

Blood vessel formation is essential for tumor growth, as it depends on a proper nutrient and oxygen supply. VEGF is not only a master regulator of angiogenesis [10] and the immune response [73] but can also stimulate tumor cell proliferation in breast cancer [74] and acute myeloid leukemia models [75]. It remains poorly understood whether endothelial-secreted VEGF is involved in these processes. However, there is evidence that the endothelial expression of VEGF-C promotes leukemic cell survival and proliferation by activating VEGF receptor-3 on cancer cells [69]. Endothelin is an angiocrine factor which regulates the vascular tone under physiological conditions. However, it can also promote ovarian carcinoma cell proliferation [35,36]. Moreover, endothelial Delta-like 4 (DLL4), which is important for controlling blood vessels formation through activating Notch1 receptors [76], can bind to Notch3 receptors on tumor cells to support their survival [32]. Interestingly, ECs can also release angiocrine factors such as Slit2, which inhibit cancer cell proliferation and motility [64]. Future work will have to assess to which extent the release of angiocrine factors contributes to cancer cell proliferation and whether this could be targeted to enhance chemotherapy.

#### 3.2.2. Cancer Stem Cells (CSCs) and Chemoresistance

Stem cells are characterized by the capacity to self-renew and the ability to differentiate into diverse specialized cell types. It is assumed that a subpopulation of stem-like cells within tumors, known as cancer stem cells (CSCs), which exhibit characteristics of both stem cells and cancer cells, have the ability to seed tumors when transplanted into an animal host. There is increasing evidence suggesting that CSCs are resistant to conventional chemotherapy and radiation treatment. The adoption of the CSC phenotypes appears to depend on signals from neighboring cells which are capable of forming a stem cell niche [77]. Intriguingly, under physiological conditions, ECs are responsible for the self-renewal and repopulation of hematopoietic cells, for instance through Notch signaling, fibroblast growth factor-4 (FGF-4) and CXCL12 cytokine expression [78,79], indicating that at least in the bone marrow ECs are capable of forming stem cell niches. It is therefore not surprising that ECs, through their membrane-bound or angiocrine secreted factors, also play a role in the acquisition of the CSC phenotype. Indeed, tumor ECs can induce the expression of genes involved in the CSC phenotype [80]. Similar to its functions within hematopoietic stem cell niches, the expression of Notch ligands such as Jagged-1 on ECs promotes the CSC phenotype in colorectal [49] and breast cancer [50,51] by activating Notch receptors. Moreover, ECs promote a stem-like phenotype of glioma cells through the secretion of Shh and activating the Hedgehog pathway in cancer cells [63] or by secreting the basic fibroblast growth factor (bFGF) [38]. VEGF can be released by ECs to promote the CSC phenotype through its receptor Neuropilin-1 on skin cancer cells [70]. There is also evidence that EC-derived cytokines are involved in adopting a CSC phenotype. The most prominent examples are IL-8 and CXCL12 in glioblastoma [30] and gastric cancer [31]. Furthermore, IL-6 secreted by tumor ECs is responsible for the generation of a small sub-population of CSCs in head and neck squamous cell carcinomas [43].

CSCs are associated with chemoresistance [77], and in mouse models, cancer treatment with chemotherapy upregulated IGF1 expression in tumor ECs, which activated IGF1 receptors on cancer cells, making them resistant to chemotherapy [42]. As such, there is evidence that angiocrine factors contribute to the adoption of the CSC phenotype and potentially also to chemoresistance. Future studies are needed to elucidate whether these factors are suitable for drug targeting and how this would affect cancer progression.

#### 3.2.3. Epithelial to Mesenchymal Transition

The epithelial to mesenchymal transition (EMT) of cancer cells describes a highly dynamic and reversible process in which cancer cells of epithelial origin lose some of their typical features, like cell-cell and cell-matrix adhesion, and gain migratory and invasive properties which are typical of mesenchymal cells. This is associated with profound changes in gene transcription. The EMT of cancer cells increases their metastatic potential and resistance toward chemotherapy [81]. Tumor ECs are involved in providing factors that influence EMT. For example, the EC-secreted EGF induces EMT transition in head and neck cancer cells [34]. Furthermore, ECs enhance EMT, breast cancer cell migration, invasion and metastasis by the release of the plasminogen activator inhibitor-1 (PAI-1) and the chemokine CCL5 [26]. At this moment, it is not yet clear whether angiocrine factors are involved in the EMT of a variety of cancer cells or only under certain conditions and cancer entities.

#### 3.2.4. Invasion and Metastasis

Cancer invasion relies on the detachment from the basal membrane, remodeling of cell-cell and cell-matrix adhesions and remodeling of the extracellular matrix. These processes are also required for invasion into blood vessels (intravasation), which is a crucial step in the metastatic cascade. Invasion and metastasis are facilitated by the EMT of cancer cells [82]. Tumor ECs are involved in EMT (as described above) and play an active role in different aspects of invasion and metastasis.

##### Extracellular Matrix (ECM) Remodeling

Changes in the ECM influence the motility and invasion of tumor cells. Matrix metalloproteinases (MMPs) are a family of proteinases that degrade the components of the ECM and thus play a major role in ECM remodeling. MMP activity is inhibited by specific tissue inhibitors of metalloproteinases (TIMPs). Several cell types, including ECs, express MMPs and TIMPs in a tumor mass [48]. ECs can influence ECM remodeling either by the expression of MMPs such as MMP2 and MMP9 or by the release of cytokines like CCL2, IL-8 and CXCL16, which act in a paracrine manner by upregulating the expression of MMPs in other cell types, such as tumor cells [25]. Furthermore, endothelial DLL4-mediated Notch signaling supports tumor cell invasion due to an increased MMP-9 expression by ECs [33]. Interestingly, tumor ECs from metastatic tumors showed a higher invasive potential than tumor ECs from non-metastatic tumors, and this has been linked with higher expression levels of gelatinase/collagenase IV, MMP-2 and MMP-9 [83].

Laminins are high-molecular weight glycoproteins, which are important components of the ECM and the basal membrane. Changes in laminin expression patterns are implicated in tumor cell migration and invasion. ECs secrete laminins, and this can facilitate the migration of melanoma cells [46]. In renal cell carcinoma, laminin-α4 is highly expressed in tumor blood vessels, and this correlates with a poor prognosis [47].

In summary, the presented studies indicate that ECs are involved in ECM remodeling, but very little is known of the extent to which this is causally linked to tumor progression.

##### Transendothelial Migration

Metastasis requires that tumor cells enter blood or lymph vessels (intravasation), to be transported to distant sites where they again need to cross the vessel wall (extravasation). Both transmigration steps are facilitated by the binding of tumor cells to endothelial adhesion molecules. Therefore, changes in the expression levels of vascular adhesion molecules influence the efficiency of transmigration and metastasis. For example, E-selectin, which under physiological conditions is required for leukocyte adhesion to ECs, can also bind certain tumor cells [57] and thereby promote transendothelial cell migration [56,58] or the homing of circulating tumor cells in the liver [59,60]. The activation of the endothelium by inflammatory or cancer-derived factors can lead to the shedding of E-selectin into the bloodstream. This interacts with CD44 on circulating tumor cells and promotes their adhesion and migration strength [61]. E-selectin also acts as a homing receptor in the hematogenous dissemination of lung [84], prostate [85] and breast cancer [86]. The expression of E-selectin on blood vessels in the bone promotes the mesenchymal-to-epithelial transition of disseminated tumor cells and the activation of Wnt signaling, which drives the stemness of cancer cells. This results in increased bone metastasis [87]. In this regard, E-selectin inhibition may interfere with the homing of metastatic cancer cells in the lung [88] or with the survival of myeloid leukemia cells within the vascular niche [89].

Besides E-selectin, vascular adhesion molecule-1 (VCAM1) is crucial for leukocyte and tumor cell transmigration. VCAM1 is expressed by ECs and bound by tumor cells expressing integrin alpha4beta1 (VLA4). VCAM1 expression increases with inflammatory stimuli. This increases the migration of melanoma cells across activated EC layers [72] and promotes lung colonization. Endothelial Notch1 signaling upregulates VCAM1 expression, which promotes the adhesion of tumor cells to the endothelium, extravasation and lung colonization, as shown by using VCAM1-blocking antibodies [52].

Likewise, intercellular adhesion molecule-1 (ICAM1) expression on ECs plays a role in the adhesion of lung carcinoma to ECs [41], promoting the invasion and metastasis of breast cancer cells [90] and liver metastasis of colorectal cancer cells [39,40]. ICAM1 can also be shedded from the endothelium into the blood stream and interact with cancer cells to enhance their pro-metastatic potential [91,92]. Interestingly, anti-ICAM1 treatment has been proposed to interfere with tumor progression in multiple myeloma [93,94], lung [95] and breast cancer [96], showing that the targeting of angiocrine factors might be a valid therapeutic option. Furthermore, the endothelial CCR2 signaling induced by colon carcinoma cells facilitates extravasation due to an increase in vascular permeability [97].

Not only membrane-bound adhesion molecules, but also EC-derived cytokines are involved in promoting cancer cell transmigration. For instance, the angiocrine factors CXCL1 and CXCL8 induce tumor cell invasion [28]. Both chemokines have also been described to enhance the transmigration and invasiveness of different cancer cell lines in 3-D collagen fiber matrix assays [29]. The angiocrine factor CCL5 promotes the downregulation of the androgen receptor (AR) in tumor cells, which accelerates the disassembly of focal adhesions, enhancing prostate cancer invasion. Hence, the inhibition of CCL5/CCR5 signaling decreased metastasis in orthotopic mouse models [27].

Tumor ECs can promote invasion and metastasis also by other factors apart from adhesion molecules and cytokines. Biglycan is a small proteoglycan, whose activity triggers tumor cell migration by nuclear factor-κB and extracellular signal-regulated kinase 1/2 signaling. Biglycan expression was found to be upregulated only in tumor ECs of highly metastatic tumors [24]. Notably, ECs can also regulate the transendothelial migration of cancer cells through the endothelial ligand EphrinA1, which binds to Ephrin-Type-A receptor 2 (EPHA2) on cancer cells [37]. Very recently, it was shown that disseminated cancer cells secrete RNA to trigger Slit2 secretion, which promotes cancer cell migration, intravasation and metastasis [65].

Lastly, angiopoietin-2 (Ang2), which is stored in Weibel Palade bodies of ECs, is a highly interesting angiocrine factor controlling tumor progression. Ang2 levels have been related to poor prognosis, for example in melanoma [98]. Ang2 blockage showed a reduction of tumor progression, angiogenesis and metastasis [20,22,99,100,101,102]. Moreover, the blockage of the angiopoietin receptor Tie1 strongly impeded transmigration and metastasis [103]. In fact, a combination of Ang2 and Tie1 blockage improves antiangiogenic therapy [104]. Recently, a landmark study demonstrated that the Ang–Tie pathway is crucial in controlling lymphatic metastasis and that this can be prevented by antibody treatment in mouse models [21].

In summary, it became clear that several membrane-bound and soluble angiocrine factors promote the transmigration and metastasis of tumor cells. Some of these factors could even be targeted by drugs, and this resulted in promising results at least in animal models. However, we still lack knowledge about the detailed mechanism whereby cancer cells, or other cells within the tumor stroma, or even systemic factors, influence ECs so that these express higher levels of such metastasis-promoting angiocrine factors. Such understanding will however be key for future translation into clinical studies.

#### 3.2.5. Pre-Metastatic Niche

When tumor cells travel through the blood stream, they first interact with ECs during colonization at distant sites. There is increasing evidence that tumor-derived signals change transcriptional programs in ECs and immune cells at distant sites to facilitate later metastatic spreading [105]. This concept of a pre-metastatic niche is still under debate. However, there are clear hints that ECs contribute to the homing and survival of circulating tumor cells [106,107,108]. For instance, disseminated breast tumor cells reside in close proximity to ECs at distant sites and endothelial-derived thrombospondin-1 (TSP-1) maintain breast cancer cell quiescence for a long period. However, upon activation of these ECs and the subsequent angiogenic sprouting, this angiocrine effect is suppressed, allowing the formation of tumor nodules [68]. In addition, the presence of a subcutaneous tumor in mice can over-activate Notch1 signaling, not only in ECs within the tumor, but also at sites as distant as the lungs. This increases endothelial VCAM1 expression, which facilitates cancer cell homing and the formation of secondary tumors [52]. During ageing, platelet-derived growth factor (PDGF)-B expression levels increase, regulating the quiescence of the dormant disseminated tumor cells and influencing therapy resistance in bone metastasis. Therefore, bone metastasis probability increases with age [109].

These studies indicate the importance of the ECs also for the preconditioning of the metastatic niche. However, much more needs to be learnt about the factors by which a primary tumor communicates with ECs at distant sites.

### 3.3. Angiocrine Control of the Immune Response

The infiltration of immune cells into the TME is of striking importance for tumor progression and metastasis, as cancer cells often generate an immunosuppressive milieu [110,111]. Leucocyte infiltration into the tumor requires interactions with several EC receptors, selectins, ICAM1 and VCAM1 [112]. As described above, some of these molecules can also be used by tumor cells during transmigration.

#### 3.3.1. Myeloid-Derived Suppressor Cells (MDSCs) and Tumor-Associated Macrophages (TAMs)

Myeloid derived suppressor cells (MDSCs) comprise a heterogeneous group of immunosuppressive immature myeloid cells which inhibit T cell and natural killer (NK) cell activity. In physiological conditions, myeloid progenitor cells differentiate into mature myeloid cells like macrophages and dendritic cells (DC), whereas, in pathological conditions like cancer, the differentiation of myeloid progenitor cells is impaired, leading to the formation of MDSCs [113]. The majority of tumor-associated macrophages (TAMs) also has an anti-inflammatory phenotype which promotes tumor progression [114]. Several chemotactic cytokines are known to recruit MDSCs and TAMs into the microenvironment [114,115,116]. Here, we only focus on angiocrine factors and how these contribute to immune cell infiltration into tumors.

Angiopoietin-2 can be secreted by tumor ECs, which leads to an autocrine activation of STAT3 signaling with secretion of CCL2 and higher expression of ICAM1 [23]. These factors promote the infiltration of CCR2-expressing monocytes into the TME [115,117]. Moreover, sustained Notch1 signaling in ECs, induced by cancer cells, drives cytokine and VCAM1 expression and the infiltration with myeloid cells, which facilitates metastasis [52]. Furthermore, E-selectin expression on ECs is required for myeloid cell infiltration in different mouse tumor models [62]. Again, this is promoted by the interaction of tumor cells with ECs, which leads to a higher E-selectin expression on tumor endothelium [62,118].

Lastly, there is increasing evidence that ECs can also impact on the polarization of recruited immune cells. In glioblastoma, the tumor endothelium was shown to be the main source of IL-6 secretion, which contributes to the adoption of an anti-inflammatory and pro-tumorigenic macrophage phenotype [44]. As such, there is still little knowledge on how ECs affect the polarization of tumor-infiltrating myeloid cells and how this contributes to immunosuppression.

#### 3.3.2. Tumor-Infiltrating Lymphocytes (TILs)

The cytotoxic function of tumor-infiltrating lymphocytes (TILs) is often impaired, for example by the activation of programmed cell death-1 (PD-1) and cytotoxic T-lymphocyte-associated protein-4 (CTLA4) receptors on T cells [53,54,55]. The expression of the respective ligands not only on cancer cells but also on other cells of the TME contributes to immunosuppression [119]. The role of ECs in this regard is still poorly understood. However, pre-clinal models [120] as well as recent clinical trials in hepatocellular carcinoma and non-small cell lung carcinoma demonstrated that antiangiogenic therapy targeting VEGF synergizes with immunotherapy targeting the PD-1/PD-L1 axis [121,122]. It will be of outstanding importance to unravel the mechanism responsible for this. In addition, there is evidence for the angiocrine control of T cell activity in tumors. Tumor ECs can secrete Inf-γ [45] and VEGF-A [71], which influence T cell responses. In addition, the induction of T cell immunoglobulin and mucin domain-containing molecule 3 (TIM-3) expression on ECs in lymphoma inhibits CD4+ T helper cell activity by the activation of the IL-6/STAT3 signaling pathway [66,67]. As such, there is some evidence that ECs play a role in adapting immune responses in cancer.

## 4. Future Perspectives

During the last years it became clear that tumor ECs play a major role in tumor progression and metastasis. Thereby, ECs are not only needed as building blocks for new blood vessels, but also as a rich source of angiocrine factors acting on cancer cells and other cells of the TME. Targeting specific angiocrine factors which orchestrate cancer proliferation, stemness, EMT, invasion and immunosuppression may improve cancer therapy. However, to achieve this, it will be of the utmost importance to unravel the mechanisms involved in the activation of angiocrine signatures, to elucidate their detailed mode of action within the TME and determine their exact contribution to tumor progression.

Anti-angiogenic therapies in combination with immune-therapy showed promising outcomes for certain cancer entities [11]. In mouse models, several approaches blocking angiogenic factors such as VEGF or Ang2 led to synergistic effects with immunotherapy. This not only normalizes the tumor vasculature to a certain degree, but also improves the infiltration with immune cells attacking cancer cells [123,124,125]. The IMBrave phase 3 trial demonstrated that such therapy can prolong the overall survival in patients with unresectable hepatocellular carcinoma when compared to treatment with Sorafenib [19]. It can be speculated that this effect is at least in part mediated by angiocrine factors. It will be of the utmost importance to unravel the angiocrine landscape in primary tumors and metastases in a systemic manner during the complete course of tumor progression in a large variety of cancer entities. These results should lead to the identification of key angiocrine factors. Further analysis will elucidate their functions in great detail within the tumor microenvironment and the metastatic niche. This should finally pave the way for initiating clinical trials to specifically interfere with angiocrine factors promoting tumor progression and metastasis.

## Figures and Tables

**Figure 1 cancers-13-02610-f001:**
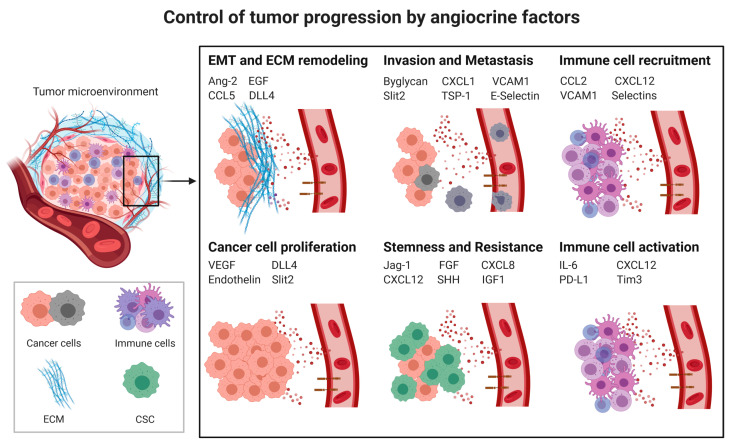
Control of tumor progression by angiocrine factors; CSC: cancer stem cells; EMT: Epithelial to mesenchymal transition; ECM: Extracellular matrix; Ang2: Angiopoietin-2, CCL2: C-C Motif Chemokine Ligand 2, CCL5: CC-chemokine ligand 5, CXCL1: C-X-C Motif Chemokine Ligand 1, CXCL8: C-X-C Motif Chemokine Ligand 8, CXCL12: C-X-C Motif Chemokine Ligand 12, DLL4: Delta-like 4, EGF: Epidermal Growth Factor, FGF: Fibroblast Growth Factor, ICAM1: Intercellular Adhesion Molecule 1, IGF1: Insulin-like growth factor 1, IL6: Interleukin-6, PD-L1: Programmed death-ligand 1, Shh: Sonic Hedgehog, Tim-3: T-cell immunoglobulin and mucin-domain containing-3, TSP-1: Thrombospondine-1, VEGF: Vascular endothelial growth factor, VCAM1: Vascular cell adhesion protein 1.

**Table 1 cancers-13-02610-t001:** List of tumor-induced angiocrine factors and their described functions.

Angiocrine Factor ^1^	Function	Reference
Ang-2	Invasion/metastasis; myeloid cell recruitment	[20,21,22,23]
Biglycan	TCs invasion/metastasis	[24]
CCL2/MCP-1	Myeloid cells recruitment; tumor cell extravasation	[25]
CCL5	EMT; invasion	[26,27]
CXCL1	Invasion	[28,29]
CXCL8/IL8	Stemness; invasion; myeloid cell recruitment; lymphocyte phenotype	[25,28,30]
CXCL12/SDF-1	Stemness	[31]
DLL4	Cancer cell proliferation; invasion	[32,33]
EGF	EMT	[34]
Endothelin	Cancer cell proliferation	[35,36]
EphrinA1	Transendothelial migration	[37]
FGF	Stemness	[38]
ICAM1	Myeloid cell recruitment/adhesion; lymphocyte phenotype	[39,40,41]
IGF1	Stemness	[42]
IL6	Myeloid cell recruitment/activation; stemness; lymphocyte phenotype	[43,44]
Inf- γ	Lymphocyte phenotype; T cell inhibition	[45]
Laminin	ECM; invasion/metastasis	[46,47]
MMPs	ECM	[48]
Notch signaling/Jag1	Stemness; CSCs; immune cell recruitment; metastasis	[49,50,51,52]
PD-L1	T cell inhibition	[53,54,55]
Selectin	Myeloid cell recruitment; immune cell and tumor cell transmigration/extravasation	[56,57,58,59,60,61,62]
Shh	Stemness	[63]
Slit2	Cancer cell proliferation, intravasation, metastasis	[64,65]
Tim-3	T cell inhibition	[66,67]
TSP-1	Pre-metastatic niche	[68]
VEGF family	Angiogenesis; cancer cell proliferation; stemness; myeloid cell recruitment; lymphocyte phenotype	[69,70,71]
VCAM1	Myeloid cell recruitment/adhesion; metastasis	[23,52,72]

^1^ Ang2: Angiopoietin-2, CCL2: C-C Motif Chemokine Ligand 2, CCL5: CC-chemokine ligand 5, CXCL1: C-X-C Motif Chemokine Ligand 1, CXCL8: C-X-C Motif Chemokine Ligand 8, CXCL12: C-X-C Motif Chemokine Ligand 12, DLL4: Delta-like 4, EGF: Epidermal Growth Factor, FGF: Fibroblast Growth Factor, ICAM1: Intercellular Adhesion Molecule 1, IGF1: Insulin-like growth factor 1, IL6: Interleukin-6, Inf-γ: Interferon-γ, MMPs: Matrix metalloproteinases, ECM: Extracellular matrix, PD-L1: Programmed death-ligand 1, Shh: Sonic Hedgehog, Tim-3: T-cell immunoglobulin and mucin-domain containing-3 (TIM-3), TSP-1: Thrombospondine-1, VEGF: Vascular endothelial growth factor, VCAM1: Vascular cell adhesion protein 1.

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
