# Peer review of "Control of Tumor Progression by Angiocrine Factors"

_cancers, 2021, doi:10.3390/cancers13112610_

Round 1

Reviewer 1 Report

The authors have produced a well written concise review of the latest developments in an exciting and promising line of research. 

They provide a comprehensive description of the major findings regarding angiocrine function in tumour progression, giving the reader a well referenced update or introduction to this theme.

I have no doubts in recommending its publication in the current form.

Author Response

The authors have produced a well written concise review of the latest developments in an exciting and promising line of research. 

They provide a comprehensive description of the major findings regarding angiocrine function in tumour progression, giving the reader a well referenced update or introduction to this theme.

I have no doubts in recommending its publication in the current form.

We thank the reviewer for the very positive evaluation of our manuscript.

Reviewer 2 Report

The field of angiocrine research is fascinating and the authors did a commendable job in presenting a status quo of the vascular oncology research. Please consider the below-listed suggestions for further advancing the manuscript.

  1. The future perspectives section is quite short and lacks a future roadmap to propel angiocrine research. To be truthful, this section doesn’t do justice to the rest of the article. I would strongly suggest elaborating this section and discuss, especially in light of the IMBrave trial, the potential impact of vascular-targeting therapies on enhancing the efficacy of immunotherapies and on modulating host response in general to tumor therapy.
  2. Authors should refrain from making a generic statement such as “… dramatically improves the outcome in patient with hepatocellular carcinoma” (line 78-80). They should always mention what is this increase over. I would even suggest mentioning the OS data from the IMBrave trial and discussing it more in detail as this form the basis of further discussions in this review.
  3. Authors should consider adding a section on Endothelial-Mesenchymal transition as well as vascular mimicry to further discuss the non-angiogenic role of EC in supporting tumor progression.
  4. It would make the review more appealing if authors can include an additional figure, potentially showing the molecular interactions of EC with different micro-environmental cells rather than in Table 1.
  5. There are several landmark papers published during the past years that authors have missed. They should consider discussing the following papers in the respective sections:
    1. Tavora et al., Nature 2020 (Section 3.2)
    2. Esposito et al, Nature Cell Biology 2019 (Section 3.2)
    3. Kloepper et al, PNAS 2016; Peterson et al, PNAS 2016, Schmittnaegel et al, Science Translational Medicine 2017 (Section 3.3)
  6. Authors need to carefully read their manuscript for editorial mistakes. A few are highlighted as follows:
    1. Line 44: secreted factor(s)
    2. Line 169: associated with rather than by
    3. Line 180: invasion into blood vessels is intravasation, not extravasation
    4. Line 214: E-selectin, not selecting
    5. Line 225-226: I guess it should be extravasation instead of intravasation
    6. Line 249: It should be through rather than throw
    7. Line 286 should be rephrased

Author Response

The field of angiocrine research is fascinating and the authors did a commendable job in presenting a status quo of the vascular oncology research. Please consider the below-listed suggestions for further advancing the manuscript.

We thank the reviewer for her/his very positive evaluation of our manuscript. We have changed the manuscript according to her/his comments (see below) and are confident that this has substantially helped to further improve the quality.

The future perspectives section is quite short and lacks a future roadmap to propel angiocrine research. To be truthful, this section doesn’t do justice to the rest of the article. I would strongly suggest elaborating this section and discuss, especially in light of the IMBrave trial, the potential impact of vascular-targeting therapies on enhancing the efficacy of immunotherapies and on modulating host response in general to tumor therapy.

We have extended this section and further discussed the IMBrave trial. We also now mention that it will be of utmost importance to unravel the mechanism how anti-angiogenic therapy acts in synergy with immunotherapy and whether angiocrine factors are involved. We also would like to stress that this article is NOT about angiogenesis but about angiocrine mechanisms (angiogenesis-independent) and there is still not so much knowledge about it. 

Authors should refrain from making a generic statement such as “… dramatically improves the outcome in patient with hepatocellular carcinoma” (line 78-80). They should always mention what is this increase over. I would even suggest mentioning the OS data from the IMBrave trial and discussing it more in detail as this form the basis of further discussions in this review.

We agree and we have changed this accordingly.

Authors should consider adding a section on Endothelial-Mesenchymal transition as well as vascular mimicry to further discuss the non-angiogenic role of EC in supporting tumor progression.

This review article is exclusively about angiocrine mechanisms in cancer. To our knowledge it is not yet clear whether angiocrine factors are involved in EndoMT or vascular mimicry. Therefore, we did not mention these two processes.

It would make the review more appealing if authors can include an additional figure, potentially showing the molecular interactions of EC with different micro-environmental cells rather than in Table 1.

We have changed figure 1 accordingly. Nevertheless, we would like to keep Table 1 as it gives a good overview and the direct links to references. 

There are several landmark papers published during the past years that authors have missed. They should consider discussing the following papers in the respective sections:

Tavora et al., Nature 2020 (Section 3.2)

We fully agree and apologize for not mentioning this before. We added this information in Section 3.2 "Transendothelial Migration"

Esposito et al, Nature Cell Biology 2019 (Section 3.2)

We fully agree and apologize for not mentioning this before. We added this information in Section 3.2 "Transendothelial Migration"

Kloepper et al, PNAS 2016; Peterson et al, PNAS 2016, Schmittnaegel et al, Science Translational Medicine 2017 (Section 3.3)

We agree that the combination of anti-angiogenic drugs in combination with immunotherapy is a big success at least in mouse models as well as in a few clinical trials as the IMBrave in HCC. This review article is exclusively about angiocrine mechanisms in cancer. And to our knowledge it is not yet clear whether the interplay of endothelial cells with immune cells is mediated through angiocrine factors. Therefore we did not cite these articles.  However, we fully agree with the importance of this topic and therefore we have expanded the Future Perspectives section and added this information there and cited those papers. 

Authors need to carefully read their manuscript for editorial mistakes. A few are highlighted as follows:

Line 44: secreted factor(s)

done

Line 169: associated with rather than by

done

Line 180: invasion into blood vessels is intravasation, not extravasation

done

Line 214: E-selectin, not selecting

done

Line 225-226: I guess it should be extravasation instead of intravasation

done

Line 249: It should be through rather than throw

done

Line 286 should be rephrased

done